# PEAKNOVO: TOWARDS THE ROBUST *De Novo* PEPTIDE SEQUENCING FOR SPECTRAL PEAKS

## ABSTRACT

*De novo* peptide sequencing aims to infer the corresponding peptide sequence from a mass spectrum, which is a fundamental task in proteomics. However, missing peaks are common in mass spectrometry-based *de novo* peptide sequencing and significantly influence the accuracy of peptide reconstruction. Existing methods mainly rely on the original mass-to-charge ratio and intensity of peaks, but fail to reconstruct a complete peptide sequence when key peaks are missing. To address this issue, we propose PeakNovo, a *de novo* peptide sequencing method that integrates candidate mass spectra search with masked self-distillation. To enhance the robustness of the mass spectrometry (MS) encoder in handling missing peaks, the Masked Self-distillation module feeds a masked local spectrum into the student branch and a complete global spectrum into the teacher branch for alignment. Enabling the encoder to learn consistent representations from local to global views, mitigating the effect of missing peaks. Meanwhile, PeakNovo employs an MS Fusion module to retrieve a set of candidate spectra similar to the input spectrum from the database. These candidate spectra are fused with the representation of the input spectrum to provide supplementary information for potentially missing peaks. Experiments on benchmark datasets show that PeakNovo consistently outperforms existing methods, achieving state-of-the-art performance. Our code is available at the anonymous link: https://anonymous.4open.science/r/PeakNovo-5AE5/README.md.

## 1 INTRODUCTION

Proteomics studies the composition, expression levels, and dynamic changes of all proteins in cells, tissues, or organisms. It investigates protein functions, interactions, and regulatory mechanisms in biological processes. Environmental factors and cellular states influence protein expression, and proteins often undergo various post-translational modifications. These characteristics highlight the importance of proteomics in studying biological functions and disease mechanisms (Lin et al., 2020; Uzozie & Aebersold, 2018). With technological advancements, proteomics has been widely applied in biomarker discovery, drug target identification, and systems biology research. Tandem mass spectrometry (MS/MS), as the only high-throughput method, is the core analytical technique in proteomics (Aebersold & Mann, 2003). It identifies and quantifies proteins in complex biological samples by ionizing peptides, separating and detecting them based on the mass-to-charge ratio (*m/z*) (Zhang et al., 2013).

The core of protein identification is peptide sequencing, which infers the peptide corresponding to the input mass spectrum through computational algorithms. Currently, there are two main approaches to this task: database search and *de novo* peptide sequencing (Nesvizhskii, 2010; Griss, 2016). Database search compares the observed spectra with theoretical spectra generated from peptide sequences in a reference database, and selects the peptide with the highest match score as the output. Commonly used database search tools include SEQUEST (Eng et al., 1994), pFind (Li et al., 2005), MSFragger (Kong et al., 2017), and Open-pFind (Sun et al., 2019). However, its reliance on known peptide sequences limits its ability to identify peptides absent from the database (Ma, 2015; Ma et al., 2003). In contrast, *de novo* peptide sequencing predicts peptides directly from the input spectrum without relying on a reference database, making it suitable for identifying previously unknown peptides.

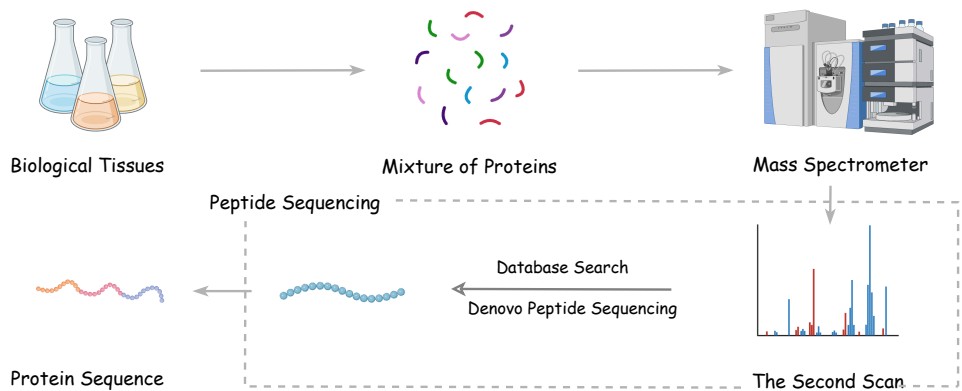

Figure 1: The workflow of protein identification using a mass spectrometer.

Most *de novo* peptide sequencing models adopt an encoder-decoder architecture (Tran et al., 2017a; Qiao et al., 2021; Yilmaz et al., 2022; Xia et al., 2024b;a). The MS encoder converts the input mass spectrum into hidden representations, which are then passed into the peptide decoder to autoregressively generate the corresponding amino acid sequence. Due to the fragmentation process and the detection mechanism of mass spectrometers, the observed spectra often suffer from missing peaks. The absence of critical peaks degrades the encoder input quality, thereby reducing the accuracy of peptide reconstruction (Zhou et al., 2024).

To address this issue, we propose PeakNovo, a robust *de novo* peptide sequencing method designed to handle missing peaks. The model consists of two components: a Masked Self-distillation module and an MS Fusion module. In the Masked Self-distillation module, a partially masked spectrum is used to simulate the missing peaks scenario. The masked and complete spectra are encoded by the MS encoder and then passed to the student and teacher branches for alignment. This encourages the encoder to learn consistent representations from local to global views, reducing the impact of missing peaks during encoding. In the MS Fusion module, candidate spectra are retrieved from a database using precursor mass and spectral similarity as constraints. These candidates are fused with the input representation to provide external evidence for potentially missing peaks.

We highlight the core contributions of this work as follows:

- *Motivation:* We propose PeakNovo, a novel *de novo* sequencing method that integrates masked self-distillation and candidate spectrum retrieval, to alleviate the problem of missing fragment peaks.
- *Algorithm:* PeakNovo enhances the MS encoder both internally and externally: it learns consistent representations from local to global views via masked self-distillation, and incorporates complementary spectral evidence through candidate spectrum fusion.
- *Experiment:* Experimental results on multiple public datasets show that PeakNovo consistently outperforms existing state-of-the-art methods, validating the effectiveness of its design.

## 2 PRELIMINARY

### 2.1 BACKGROUND

Mass spectrometry (MS) based proteomics is a powerful technique for characterizing proteins in biological tissues, enabling comprehensive analysis of their identities, structures, and functional properties. As illustrated in Figure 1, the overall analysis consists of several key steps. The workflow begins with protein extraction from biological tissues via cell lysis and centrifugation, yielding a heterogeneous protein mixture. This crude extract contains all native proteins without prior separation. The proteins are enzymatically digested (typically with trypsin) into peptides, which are more amenable to MS analysis due to their smaller size and uniform charge distribution. The resulting

peptides are then analyzed using a mass spectrometer. The peptides are ionized and analyzed by MS, which measures their mass-to-charge ratios and generates a mass spectrum. The spectrum contains mass information for different peptides, which is critical for subsequent analysis. In addition to the initial mass spectrum analysis, the instrument performs tandem mass spectrometry (MS/MS). This step further fragments the peptides to obtain more detailed mass spectra. The MS/MS spectra provide additional structural information that helps determine the composition and sequence of peptides. There are two common approaches for peptide sequence interpretation. Database Search compares the spectral data with a known mass spectra database to find highly matching peptide sequences. It enables quick identification of the source protein of each peptide. *De novo* peptide sequencing infers the amino acid sequence of peptides directly from the spectra without relying on any database. It is especially useful for analyzing unknown proteins in antibody sequencing (Tran et al., 2017b) or vaccine development (Mayer & Impens, 2021). Through these interpretation steps, the complete amino acid sequence can be reconstructed. Through peptide-spectrum matching and sequence assembly, the complete protein sequence can be reconstructed, allowing annotation of functional domains with statistical confidence scores.

## 2.2 RELATED WORK

Recent advancements in deep learning have led to the development of various neural network-based models for *de novo* peptide sequencing, which can be divided into traditional deep learning models and Transformer-based models. Traditional methods such as DeepNovo (Tran et al., 2017a) and PointNovo (Qiao et al., 2021) use architectures like CNNs (Alzubaidi et al., 2021) and LSTMs (Yu et al., 2019). DeepNovo converts spectra into discretized vectors processed by ion-CNN and LSTM, with decoding constrained by a knapsack algorithm. PointNovo improves input representation by using (*m/z*, intensity) pairs and a PointNet-like structure. Transformer-based models like Casanovo (Yilmaz et al., 2022) and InstaNovo (Eloff et al., 2023) take (*m/z*, intensity) pairs as inputs to the MS encoder, while InstaNovo (Eloff et al., 2023) further incorporating a precursor encoder. AdaNovo (Xia et al., 2024a) introduces a peptide decoder to model spectrum-peptide and amino acid mutual information for adaptive training, and $\pi$-HelixNovo (Yang et al., 2024) improves spectrum completeness by integrating complementary spectra to solve missing ion problems in MS2 data. SearchNovo (Xia et al., 2024b) retrieves candidate peptides and integrates them into the decoder to enhance peptide reconstruction.

## 3 METHOD

### 3.1 TASK FORMULATION

*De novo* peptide sequencing aims to infer the amino acid sequence of a peptide directly from tandem mass spectrometry (MS/MS) data, without relying on a reference database. Formally, a mass spectrum is represented as $\mathbf{x} = \{(m_i, t_i)\}_{i=1}^{M}$, where each pair $(m_i, t_i)$ denotes the mass-to-charge ratio (*m/z*) and the intensity of the $i$-th peak, and $M$ is the total number of peaks in the spectrum. The precursor ion, corresponding to the peptide selected for fragmentation, is denoted as $\mathbf{z} = (m_{\text{prec}}, c_{\text{prec}})$, where $m_{\text{prec}} \in \mathbb{R}$ is the precursor mass and $c_{\text{prec}} \in \{1, 2, \ldots, 10\}$ is its charge state. The corresponding peptide sequence is represented as $\mathbf{y} = (y_1, y_2, \ldots, y_N)$, where each $y_i$ denotes the amino acid at position $i$, and $N$ is the peptide length, which varies across peptides. The goal of *de novo* peptide sequencing is to model the conditional probability of the peptide sequence given the observed spectrum and precursor information, typically in an autoregressive manner:

$$P(\mathbf{y} \mid \mathbf{x}, \mathbf{z}; \theta) = \prod_{j=1}^{N} p(y_j \mid \mathbf{y}_{<j}, \mathbf{x}, \mathbf{z}; \theta), \tag{1}$$

where $\theta$ denotes the model parameters and $\mathbf{y}_{<j}$ denotes the prefix subsequence consisting of the first $j - 1$ amino acids. Model training typically minimizes the negative log-likelihood, that is, the cross-entropy loss over the sequence:

$$\mathcal{L}_{\text{CE}}(\theta) = -\sum_{j=1}^{N} \log p(y_j \mid \mathbf{y}_{<j}, \mathbf{x}, \mathbf{z}; \theta) \tag{2}$$

to optimize the parameters $\theta$. During inference, the model predicts the probability distribution over amino acids at each position in the sequence and typically uses heuristic search strategies, such as beam search, to generate the most probable peptide sequence hypotheses.

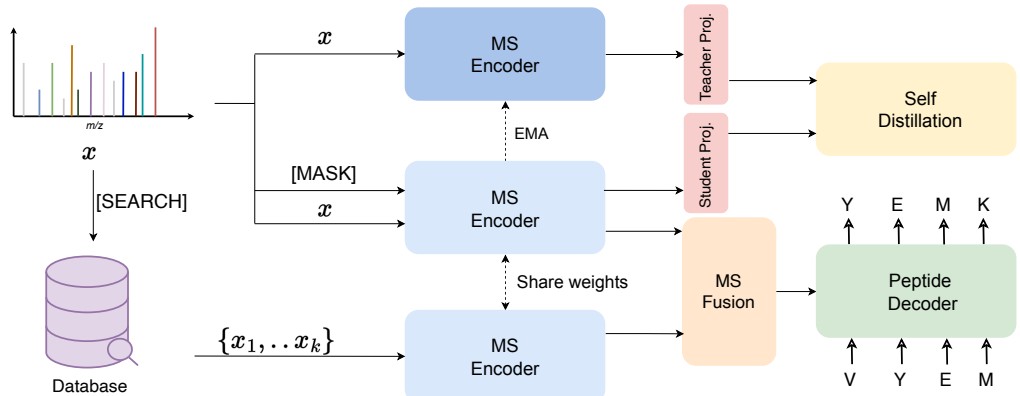

Figure 2: The overview of PeakNovo, contains Masked Self-ditillation and MS Fusion components.

## 3.2 MODEL ARCHITECTURES

As shown in Figure 2, PeakNovo consists of two main components: the Masked Self-distillation and the MS Fusion. The Masked Self-distillation component contains a Teacher MS Encoder, a Student MS Encoder, two projection layers, and a Self Distillation module. Following previous work (Yilmaz et al., 2022; Xia et al., 2024a; Yang et al., 2024), the MS Encoder is based on a transformer, and transforms $\mathbf{x} = \{(m_i, I_i)\}_{i=1}^{M}$ into a $d$-dimensional vector representation. The MS Encoder first applies a fixed sinusoidal embedding to $m$-value of each peak, calculated as follows:

$$
f_{ij} = \begin{cases} \sin\left(m_i \Big/ \left(\dfrac{m_{\max}}{m_{\min}}\left(\dfrac{m_{\min}}{2\pi}\right)^{2j/d}\right)\right), & \text{for } j \leq \dfrac{d}{2}, \\ \cos\left(m_i \Big/ \left(\dfrac{m_{\max}}{m_{\min}}\left(\dfrac{m_{\min}}{2\pi}\right)^{2j/d}\right)\right), & \text{for } j > \dfrac{d}{2}, \end{cases} \tag{3}
$$

where $f_{ij}$ denotes the $j$-th component of the embedding for the $i$-th peak, $d$ is the embedding dimension, and $m_{\max}$ and $m_{\min}$ are constants set to 10,000 and 0.001, respectively. Each peak's intensity $t_i$ is projected into a $d$-dimensional vector using a linear layer. The representation of each peak is obtained by summing the embeddings of $m_i$ and $t_i$. The embedding is then projected into the hidden representation space through the transformer. The projection layers for both the student and the teacher are implemented as single linear layers. The self-distillation module computes the self-distillation loss between the student and teacher logits' representations. The MS Fusion component consists of an MS Encoder and a cross-attention-based MS Fusion module. The Peptide Decoder is an autoregressive transformer decoder that takes the previously decoded amino acid sequence and the encoder-derived cross-attention representation as input to predict the next amino acid. The amino acid vocabulary includes 20 standard amino acids, several post-translationally modified (PTM) amino acids, and an [EOS] token that indicates the end of the decoding process.

## 3.3 MASKED SELF-DISTILLATION

Inspired by the success of self-distillation paradigm in the visual domain (Caron et al., 2021) , we adopt a masked self-distillation strategy to enhance the encoder's representation capability under missing peak conditions in MS data. Specifically, unlike the multi-view augmentations typically used in the image domain, we apply peak-level masking on peak sequences. By combining random masking with contiguous neighborhood masking, we effectively mimic real-world missing peak patterns caused by intensity thresholds or instrument-related errors. Given an input mass spectrum $\mathbf{x} = \{(m_i, t_i)\}_{i=1}^{M}$, we randomly mask a subset of the peaks based on a predefined masking ratio to obtain $\mathbf{x}'$. The masked positions are filled with [PAD] tokens to simulate the scenario of missing peaks, which frequently occurs due to noise, intensity thresholding, or incomplete measurements in real-world mass spectrometry data. The original and masked spectrum $\mathbf{x}$ and $\mathbf{x}'$ are embedded by Teacher and Student MS Encoder and their respective projection heads,, resulting in represntations $\{t_i\}_{i=1}^{M}$ and $\{s_i\}_{i=1}^{M}$. To encourage consistent representations under missing peak conditions, we

compute the distillation loss between the peak representation token as follows:

$$t'_i = \text{softmax}(\frac{t_i - c}{\tau_t}), \quad s'_i = \text{softmax}(\frac{s_i}{\tau_s}), \tag{4}$$

$$\mathcal{L}_{\text{dist}} = \frac{1}{M} \sum_{i=1}^{M} \mathcal{KL}\left(t'_i \| s'_i\right), \tag{5}$$

where $M$ is the number of mass spectrum peaks, $\mathcal{KL}(\cdot\|\cdot)$ denotes the Kullback–Leibler divergence loss, $\tau_t$ and $\tau_s$ are temperature coefficients, and $c$ is the center vector of the teacher outputs, computed as a moving average over batch-wise teacher features. The centering operation dynamically shifts the teacher's outputs to stabilize training. It helps prevent early representation collapse, where the teacher outputs degenerate into uniform or trivial solutions. The temperature coefficients smooth the probability distributions, thereby stabilizing gradient-based optimization. The parameters of the Student MS Encoder are updated via gradient descent, whereas the Teacher MS Encoder is updated using an Exponential Moving Average (EMA) mechanism:

$$\theta_t^{(e)} = m_e \cdot \theta_t^{(e-1)} + (1 - m_e) \cdot \theta_s^{(e)}, \tag{6}$$

where $\theta_s^{(e)}$ denote the parameters of the student model at step $e$, and $\theta_t^{(e)}$ denotes the parameters of the teacher model. The momentum coefficient $m_e \in [0, 1)$ controls the update speed, where a higher value leads to slower but more stable updates of the teacher model. Through masked self-distillation, the teacher model leverages the complete mass spectrum to generate informative soft labels that serve as a stable supervisory signal for the student model. This mechanism encourages the student model to align the representations of partially observed spectra with those derived from complete inputs.

Our adoption of masked self-distillation over mainstream self-supervised methods is based on the unique characteristics of MS data. Unlike reconstruction-based approaches such as masked autoencoders (MAE), MS data often contain missing peaks with unstable intensities due to noise, or instrument limitations, making reconstruction prone to overfitting on uninformative signals. Instead, we use KL divergence to align the soft outputs of teacher and student models, guiding the learning of robust and discriminative representations that are invariant to missing peaks. Furthermore, contrastive learning faces challenges in MS, such as ambiguous definitions of positive and negative pairs—different spectra may share common fragments (false negatives), while the same peptide under varying conditions may yield weak positives. In contrast, masked self-distillation allows the teacher, which sees the full spectrum, to provide stable supervision for the student, thereby avoiding the sampling noise and instability inherent in contrastive methods.

### 3.4 MS FUSION

The Masked Self-distillation component aims to improve the model's robustness under missing scenarios by enforcing representation consistency between complete and masked inputs. In contrast, the MS Fusion complements the missing fragments by retrieving and integrating information from external spectra. Specifically, given a query spectrum $\mathbf{x} = \{(m_i, t_i)\}_{i=1}^{M}$, we use MatchMS to search for top-$k$ most similar spectra $\{\mathbf{x}_1, \ldots, \mathbf{x}_k\}$ from a database, within a mass tolerance window of $\pm 20$ Da. If fewer than $k$ candidates are found, the search window is gradually expanded (e.g., by 10 Da increments) until $k$ candidates are retrieved.

Since the spectra in the candidate set potentially contain peaks that are missing in the query spectrum, we embed both the query spectrum $x$ and the candidate spectra $\{\mathbf{x}_1, \ldots, \mathbf{x}_k\}$ using the shared MS Encoder to obtain representations $z \in \mathbb{R}^{M \times d}$ for the query spectrum and $\{z_1, \ldots, z_k\}$ for the retrieved spectra, respectively. The MS Fusion module then applies a cross-attention mechanism to integrate information from the retrieved candidates into the query spectrum representation. Formally, the fused representation is computed as:

$$z' = \text{CrossAttention}(z, \{z_i\}_{i=1}^{k}, \{z_i\}_{i=1}^{k}) + z. \tag{7}$$

This mechanism allows each peak in the query spectrum to attend to informative peaks from the retrieved candidate spectra. By computing similarities between peaks and aggregating the corresponding representations, the model learns to incorporate relevant external evidence. A residual connection is used to preserve the original query information during the fusion process. The fused representation $z'$ is then used as the input to the Peptide Decoder, which performs autoregressive peptide prediction based on the enriched spectrum representation.

## 3.5 Training Objective & Model Inference

During training, we minimize the following joint loss function to optimize PeakNovo:

$$\mathcal{L}(\theta) = -\sum_{j=1}^{N} \log p(y_j \mid \mathbf{y}_{<j}, \mathbf{x}, \mathbf{z}; \theta) + \mathcal{L}_{\text{dist}}. \tag{8}$$

The first term is the autoregressive loss from the Peptide Decoder, where each amino acid $y_i$ is predicted conditioned on the previously decoded prefix sequence and the encoded spectrum representation. The second term is the masked self-distillation loss introduced earlier, which enhances the model's robustness under missing peak conditions. During inference, only the MS Fusion component is used. The query spectrum is first used to search for similar spectra from an external database. The retrieved spectra are encoded using the shared MS Encoder and fused with the query through the MS Fusion module. The fused representation $z'$ is then passed to the Peptide Decoder to generate the peptide sequence in an autoregressive manner.

## 4 Experiment

### 4.1 Datasets

Following the benchmarking protocol of *de novo* peptide sequencing from NovoBench (Zhou et al., 2024), we conducted comprehensive experiments on recent baseline models and PeakNovo using three datasets: Seven-species, Nine-species, and HC-PT. The Seven-species dataset contains low-resolution mass spectrometry data along with the corresponding peptide labels from seven different species. The model is evaluated on the yeast species, while the remaining six species are used for training. This setting simulates a practical scenario in which peptide sequences that have never been seen before need to be identified. The Nine-species dataset is the most widely used high-resolution mass spectrometry dataset and contains spectra and peptide labels from nine different species. The HC-PT dataset originates from the InstaNovo paper and contains synthetic tryptic peptides that cover all human proteins and their isoforms. This dataset contains high-resolution human-derived mass spectrometry data, and the peptide labels come from high-confidence search results using MaxQuant(Tyanova et al., 2016). It is important to note that in all three datasets, there is no overlap between peptides in the training and test sets. Traditional database search methods perform poorly under these conditions, as they are unable to identify peptide sequences that are not in the database. Detailed information about the datasets is provided in Table 1.

Table 1: The statistics of the datasets summarize the mean values of the spectra characteristics.

| Dataset | precursor $m/z$ | precursor charge | peaks num. | intensity | peptide len. |
|---|---|---|---|---|---|
| Seven-species | 719.07 | 2.42 | 466.05 | 956.17 | 15.79 |
| Nine-species | 679.68 | 2.47 | 134.91 | 175082.65 | 15.01 |
| HC-PT | 635.32 | 2.31 | 184.21 | 143363.17 | 12.53 |

### 4.2 Metrics

To evaluate *de novo* peptide sequencing performance, we calculate precision and recall at the amino acid, post-translational modification (PTM), and peptide levels. A predicted amino acid is considered correct if its mass deviates from the ground truth by less than 0.1 Da and aligns with either the prefix or suffix within 0.5 Da. The amino acid-level precision and recall are defined as $\text{Precision}^{\text{AA}} = N_{\text{match}}^{\text{aa}}/N_{\text{pred}}^{\text{aa}}$ and $\text{Recall}^{\text{AA}} = N_{\text{match}}^{\text{aa}}/N_{\text{truth}}^{\text{aa}}$, where $N_{\text{match}}^{\text{aa}}$ is the number of correctly predicted amino acids, and $N_{\text{pred}}^{\text{aa}}$ and $N_{\text{truth}}^{\text{aa}}$ represent the total predicted and ground-truth amino acids, respectively. PTM-level precision and recall are calculated as $\text{Precision}^{\text{PTM}} = N_{\text{match}}^{\text{ptm}}/N_{\text{pred}}^{\text{ptm}}$ and $\text{Recall}^{\text{PTM}} = N_{\text{match}}^{\text{ptm}}/N_{\text{truth}}^{\text{ptm}}$, where a match requires both correct localization and modification type. At the peptide level, a prediction is considered correct only if all amino acids match exactly; precision is defined as $\text{Precision}^{Pep} = N_{\text{match}}^{p}/N_{\text{all}}^{p}$, where $N_{\text{match}}^{p}$ is the number of fully correct peptides and $N_{\text{all}}^{p}$ is the total number of predictions. We use the area under the precision-recall curve (AUC) to summarize model performance across different confidence thresholds.

### 4.3 BASELINES AND EXPERIMENTAL SETTINGS

We compared PeakNovo against seven recent and representative baseline models, including Deep-Novo (Tran et al., 2017a), PointNovo (Qiao et al., 2021), Casanovo (Yilmaz et al., 2022), CasanovoV2(Yilmaz et al., 2024), $\pi$-HelixNovo (Yang et al., 2024), AdaNovo (Xia et al., 2024a), and SearchNovo (Xia et al., 2024b). We trained PeakNovo with a batch size of 32 for 30 epochs on an Nvidia A100 GPU. The learning rate was set to 0.0004, and the weight decay to $1 \times 10^{-5}$. We applied a linear warm-up schedule and used the AdamW optimizer to update model parameters. Optimal hyperparameters were selected based on the performance on the validation set. In the experiments, we treated the training set as the database to search for candidate spectra. For baseline models, we use the same hyperparameter settings reported in their original papers.

Importantly, for key peptide-level metrics such as precision and AUC-especially critical since our main goal is to reconstruct the full peptide sequence-when the missing rate reaches 60% and 80%, CasaNovoV2's peptide precision and AUC drop close to zero, while PeakNovo still maintains reconstruction capability.

### 4.4 MAIN RESULTS

**PeakNovo Outperforms State-of-the-Art Methods on Benchmark Datasets** As shown in Table 2, PeakNovo outperforms all baseline models on three datasets at both the peptide and the amino acid level. At the amino acid level on the Seven-species dataset, PeakNovo achieves a recall 4.7% higher than SearchNovo and a precision 4% higher than DeepNovo. On the Nine-species dataset, PeakNovo achieves a precision of 0.823 and a recall of 0.822, surpassing the second-best model, $\pi$-HelixNovo, by 5.8% and 6.4%, respectively. On the HC-PT dataset, PeakNovo achieves precision and recall values of 0.712, higher than CasaNovoV2's 0.694 and 0.695. Notably, at the peptide level on the Nine-species dataset, PeakNovo achieves a peptide precision of 0.630, 8% higher than SearchNovo. In terms of AUC, it exceeds the second-best CasaNovoV2 by 8.6%. On the Seven-species dataset, PeakNovo achieves a peptide recall of 0.276 and an AUC of 0.229. On the HC-PT dataset, PeakNovo achieves a peptide recall of 0.550 and an AUC of 0.550, outperforming CasaNovoV2 by 2.3% and 3.1% in these metrics, respectively.

Table 2: Empirical comparison of *de novo* sequencing models using amino acid-level and peptide-level metrics. The best and the second best are highlighted with **bold** and underline, respectively.

| Method | Amino acid-level performance | | | | | | Peptide-level performance | | | | | |
| --- | --- | --- | --- | --- | --- | --- | --- | --- | --- | --- | --- | --- |
| | Seven-species | | Nine-species | | HC-PT | | Seven-species | | Nine-species | | HC-PT | |
| | Prec. | Recall | Prec. | Recall | Prec. | Recall | Prec. | AUC | Prec. | AUC | Prec. | AUC |
| DeepNovo | 0.492 | 0.433 | 0.696 | 0.638 | 0.531 | 0.534 | 0.204 | 0.136 | 0.428 | 0.376 | 0.313 | 0.255 |
| PointNovo | 0.196 | 0.169 | 0.740 | 0.671 | 0.623 | 0.622 | 0.022 | 0.007 | 0.480 | 0.436 | 0.419 | 0.373 |
| Casanovo | 0.322 | 0.327 | 0.697 | 0.696 | 0.442 | 0.453 | 0.119 | 0.084 | 0.481 | 0.439 | 0.211 | 0.177 |
| CasanovoV2 | 0.179 | 0.181 | 0.752 | 0.750 | 0.694 | 0.695 | 0.029 | 0.017 | 0.543 | 0.514 | 0.527 | 0.489 |
| $\pi$-HelixNovo | 0.481 | 0.472 | 0.765 | 0.758 | 0.588 | 0.582 | 0.234 | 0.173 | 0.517 | 0.453 | 0.356 | 0.318 |
| AdaNovo | 0.379 | 0.385 | 0.698 | 0.709 | 0.442 | 0.451 | 0.174 | 0.135 | 0.505 | 0.469 | 0.212 | 0.178 |
| SearchNovo | 0.489 | 0.488 | 0.748 | 0.746 | 0.652 | 0.658 | 0.259 | 0.174 | 0.550 | 0.489 | 0.447 | 0.413 |
| PeakNovo | **0.532** | **0.535** | **0.823** | **0.822** | **0.712** | **0.712** | **0.276** | **0.229** | **0.630** | **0.600** | **0.550** | **0.520** |

Post-translational modification (PTM) is a mechanism through which cells fine-tune protein functions after translation . Its diversity and reversibility enable proteins to respond flexibly to internal and external environmental changes across both temporal and spatial dimensions. Therefore, PTM analysis is important in proteomics, disease mechanism studies, and drug development (Deribe et al., 2010). As illustrated in Table 3, at the PTM level on the Nine-species and HC-PT datasets, PeakNovo achieves a recall of 0.652, surpassing SearchNovo by 5.3%. PeakNovo also achieves a recall of 0.830, outperforming CasaNovoV2's 0.796. Specifically, on the Nine-species dataset, PeakNovo surpasses SearchNovo by 4% in PTM precision. On the Seven-species dataset, PeakNovo achieves a PTM recall of 0.452 and a precision of 0.522.

In addition, we observe that PeakNovo performs better on the high-resolution Nine-species and HC-PT datasets compared to the lower-resolution Seven-species dataset. This indicates that high-resolution mass spectrometry data provides more informative data and benefits both the Mask Self-distillation and MS Fusion components of PeakNovo.

Table 3: Empirical comparison of *de novo* sequencing models in terms of identifying PTMs. The best and the second best are highlighted with **bold** and underline, respectively.

| Method | PTM Recall | | | PTM Prec. | | |
|---|---|---|---|---|---|---|
| | Seven-species | Nine-species | HC-PT | Seven-species | Nine-species | HC-PT |
| DeepNovo | 0.373 | 0.529 | 0.615 | 0.391 | 0.576 | 0.626 |
| PointNovo | 0.094 | 0.546 | 0.740 | 0.117 | 0.629 | 0.676 |
| Casanovo | 0.251 | 0.566 | 0.460 | 0.360 | 0.706 | 0.501 |
| CasanovoV2 | 0.117 | 0.553 | 0.796 | 0.209 | 0.714 | 0.750 |
| $\pi$-HelixNovo | 0.366 | 0.598 | 0.667 | 0.473 | 0.680 | 0.568 |
| AdaNovo | 0.321 | 0.570 | 0.482 | 0.448 | 0.652 | 0.552 |
| SearchNovo | 0.447 | 0.599 | 0.772 | 0.472 | 0.764 | 0.715 |
| PeakNovo | **0.452** | **0.652** | **0.830** | **0.522** | **0.804** | **0.753** |

**PeakNovo Effectively Mitigates the Impact of Missing Peaks** We further evaluate PeakNovo's performance under missing peak scenarios. Using the high-resolution Nine-species dataset, we randomly dropped spectral peaks in the test set at missing rates of 20%, 40%, 60%, and 80%. We comprehensively compare PeakNovo with CasaNovoV2, a model trained on large-scale mass spectrometry data, under different missing rate conditions. As shown in Figure 3, across different missing peak ratios, PeakNovo consistently outperforms CasaNovoV2 at the peptide, amino acid, and PTM level. When the missing rate is low, PeakNovo can make full use of the available spectral information. Notably, at a 40% missing rate, PeakNovo achieves higher peptide precision than CasaNovoV2 at a 20% missing rate.

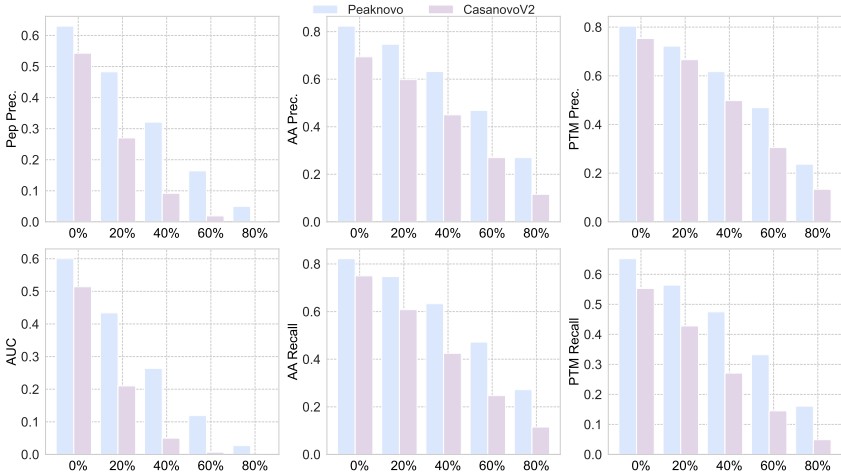

Figure 3: Performance of PeakNovo and CasanovoV2 under different missing peak ratios.

## 4.5 ABLATION STUDY

In this section, we investigate the individual contributions of the two components in PeakNovo: Masked Self-distillation (MSD) and MS Fusion (MF). We ablate four models on the Nine-species dataset: PeakNovo, PeakNovo without (w/o) MF, PeakNovo w/o MSD, and the base model without either component. As shown in Figure 4, both the MF and the MSD component outperform the base model, demonstrating the effectiveness of the two modules. PeakNovo w/o MSD performs better than PeakNovo w/o MF, indicating that external information is more important under missing peak scenarios. Furthermore, PeakNovo outperforms both PeakNovo w/o MSD and PeakNovo w/o MF, showing that enhancing the encoder's robustness and supplementing missing peak information from external sources complement each other.

**Sensitivity Analysis** In this section, we investigate the impact of the mask ratio $r$ in Masked Self-distillation and the number of candidate spectra $k$ in MS Fusion on the performance of PeakNovo. As can be observed from Figure 5, as the mask ratio increases during training, the performance of PeakNovo first improves and then declines. When the mask ratio is high, the few remaining spectral peaks prevent the model from effectively learning the consistency between local and global

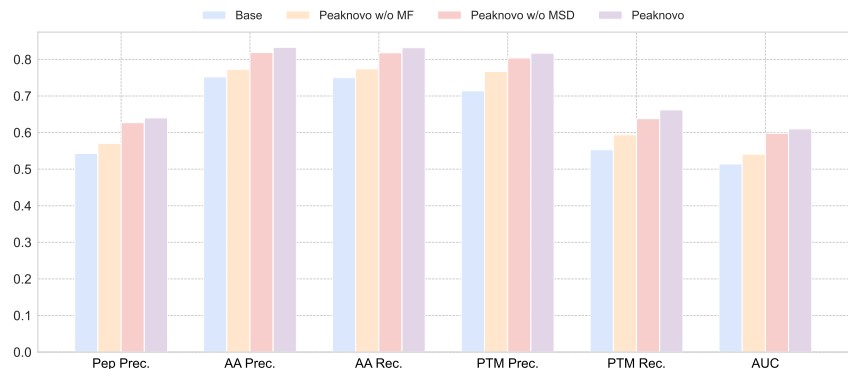

Figure 4: Ablation performance of PeakNovo on various metrics across the nine-species dataset.

representations, which in turn affects the *de novo* sequencing process. Similarly, the performance of PeakNovo with respect to $k$ shows the same trend. When $k$ becomes large, the candidate spectrum set fails to provide additional useful missing peak information and instead introduces noise.

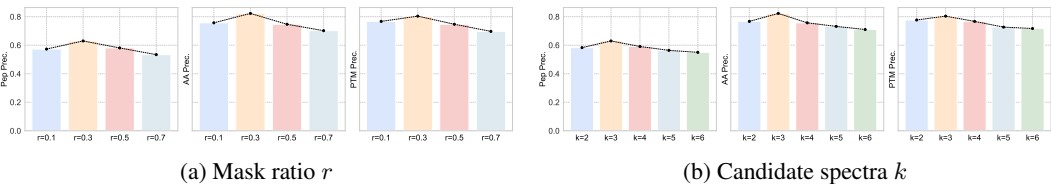

(a) Mask ratio $r$                                        (b) Candidate spectra $k$

Figure 5: Sensitivity analysis of mask ratio $r$ and the number of candidate spectra $k$.

**Case Study** To verify whether the candidate spectra used by PeakNovo can alleviate the missing peak problem, we use Pyteomics (Goloborodko et al., 2013) to annotate the $b$ and $y$ ions in the spectra. As shown in Figure 6, the reference spectrum and the query spectrum share several common peaks such as $y2$, $y4$, and $y5$. In addition, the reference spectrum contains the $y14$ peak, which is missing in the query spectrum. This enables the MS Fusion module to provide additional information and mitigate the impact of the missing peak in the original spectrum.

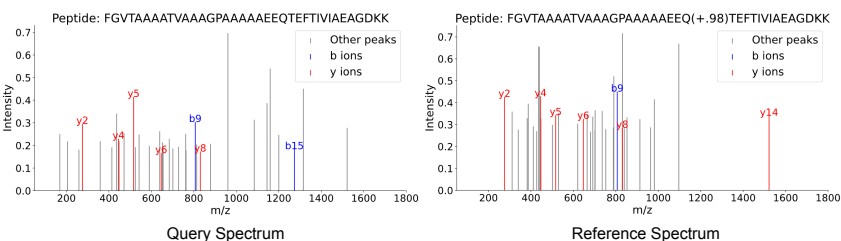

Figure 6: Comparison of query spectrum and reference spectrum annotated by Pyteomics.

## 5 CONCLUSION

In this paper, we introduce PeakNovo, a *de novo* sequencing method that integrates masked self-distillation and candidate spectrum search. The Masked Self-distillation and MS Fusion modules in PeakNovo enhance performance under missing peak scenarios by improving the MS encoder and incorporating information from external candidate spectra. Experimental results on public mass spectrometry benchmark datasets show that PeakNovo achieves state-of-the-art performance while effectively mitigating the impact of missing peaks. Future work includes exploring methods to reduce the influence of noise peaks in mass spectra, enabling broader application in proteomics analysis.

## ETHICS STATEMENT

We adhere to the ICLR Code of Ethics in all aspects of this research. All experimental data and results presented are authentic, with full transparency in methodology and analysis. We ensure that no form of plagiarism or falsification has occurred, and all sources are properly cited. In our work, we have taken steps to ensure fairness, equity, and respect for all contributors, avoiding any form of bias or discrimination. All contributors to this research are duly acknowledged for their roles, and any conflicts of interest have been disclosed. We have also respected the confidentiality and privacy of any sensitive data, ensuring compliance with relevant ethical guidelines. The peer review process has been conducted with the utmost respect for fairness and objectivity. Finally, we are committed to promoting an inclusive and open academic community and fostering a culture of collaboration and mutual respect.

## REPRODUCIBILITY STATEMENT

We are committed to ensuring the reproducibility of our results. The code and data used in this study will be made publicly available upon publication, and we provide clear instructions for their usage to enable replication of our experiments. All hyperparameters, datasets, and experimental setups are described in detail in the paper to allow others to reproduce our findings.

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

## USE OF LARGE LANGUAGE MODELS

In accordance with ICLR's disclosure requirements regarding the use of large language models (LLMs), this section specifies the role and scope of LLM usage in this work:

1. **As a writing refinement tool (limited assistance):** During manuscript preparation, LLMs were used to provide limited assistance with grammar correction, word choice, sentence restructuring, and improving overall fluency. Importantly, LLMs were not used to draft or rewrite any core scientific content, including conceptual arguments, experimental design, data analysis, results interpretation, or figure preparation. The specific prompts and representative examples of edits have been retained by the authors and can be made available for review upon request.

2. **Accountability and verification:** All text generated or modified with LLM assistance was carefully reviewed by the authors to ensure the absence of plagiarism, fabricated information, or misleading statements. The authors take full responsibility for all content in the manuscript. LLMs are not listed as authors or contributors to this work.

