# OpenReview forum: "PeakNovo: Towards the Robust De Novo Peptide Sequencing for Missing Spectral Peaks"
_ICLR.cc/2026/Conference — Submitted to ICLR 2026_

### Official Review · Reviewer_tmCS · 2025-10-27

**Soundness:** 2
**Presentation:** 3
**Contribution:** 2
**Rating:** 2
**Confidence:** 4

**Summary:**

This work proposes a new method for de novo sequencing, called PeakNovo.  The method involves two novel components: a masked self-distillation approach that is designed to help the model cope with missing peaks, and an "MS fusion" approach that makes use of spectra derived from an external database that closely resemble the query spectra.  An existing benchmark is used to show that the proposed method outperforms state-of-the-art methods.

**Strengths:**

I thought that the masked self-distillation idea was nice.  The ablation study suggests that this is working well, so this could be a contribution to the field.

Section 3.1 provides a very nice summary of the problem.

**Weaknesses:**

The idea of grabbing potentially replicate spectra from an external database for use by the model is pretty unsatisfying.  Indeed, my main critique of this work is that, although you claim to be solving the de novo sequencing problem, in practice you really aren't.  Instead, you are solving some new, closely related task where the input is a spectrum plus some (potentially huge) database of additional spectra, and the output is the amino acid sequence of the spectrum.  Where this database should come from or how to select its contents seems quite problematic.  In particular, there is a potential for a sort of leakage to occur here, since the auxiliary database may well contain multiple spectra that were generated from the same peptide that you are attempting to decode.

I found the abstract confusing because in line 24 you suddenly start talking about a database, even though you are supposedly proposing a de novo sequencing method.  It was only at line 255 (on p. 5) that I finally figured out that you are talking about a database of spectra rather than peptides.  This is very confusing, because (for whatever reason), in MS/MS analysis, a collection of peptides is called a "database" whereas a collection of spectra (used in search) is called a "library."  This database needs to be introduced and clearly explained much earlier.  In particular, it's important to indicate whether these are annotated spectra (i.e., spectra with associated peptides) or just spectra.  From the description, it sounds like it is the latter.

Given the concerns outlined above,  I was pretty shocked to see that you never even told us what spectrum database was used in the experiments reported in Section 4.4.  This makes the results impossible to reproduce, and it also makes it impossible to figure out whether there is a potential for leakage.

Minor:

Line 71: You should qualify the initial phrase a bit.  This claim only applies to recent deep learning models.

Line 107: The charge distribution is not uniform. There are many more +2 charges than +1, for example.

Line 132: InstaNovo's novelty does not lie in introducing a precursor encoder.

At around line 182, I think it would be sensible to provide a brief overview of how the model works, prior to diving into the details.  You need to explain that there are two encoders and how they related to the student and teacher modules.  As it is, the student and teacher are mentioned on line 195 without ever being introduced.

Line 215: Typos (double comma, misspelled word)

I don't understand how missing peaks can have "unstable intensities" (line 140).  A missing peak has no intensity.  I am guessing that you mean missing peaks AND unstable intensities (in the non-missing peaks).

Line 297: It's important to clarify that the data from the InstaNovo paper consists of spectra (not peptides), and that those are real spectra derived from synthesized peptide sequences.

Line 328: Doesn't Casanovo have actual version numbers associated with various releases of the software?  You should use those, rather than saying CasanovoV2.

In Section 4.3, you should make clear that you are simply using previously reported results for all methods except your own, and you should state where those results are being taken from.

**Questions:**

Why is there no weighting parameter on the two terms in the loss function (Equation 8)?

Maybe I missed it, but I don't think you ever explained what is the architecture of the self-distillation module.  What is the architecture of that module?

What database of spectra did you use?  How did you ensure that it does not contain spectra from your test set?

---

### Official Review · Reviewer_6c15 · 2025-10-27

**Soundness:** 3
**Presentation:** 3
**Contribution:** 2
**Rating:** 4
**Confidence:** 3

**Summary:**

This paper proposes PeakNovo, a robust de novo peptide sequencing method designed to address the critical issue of missing peaks in mass spectrometry (MS) data—a major limitation of existing methods that degrades peptide reconstruction accuracy. The core design of PeakNovo integrates two key modules:
（1）Masked Self-distillation: Simulates missing peak scenarios by feeding a masked local spectrum to the student branch and a complete global spectrum to the teacher branch, enabling the MS encoder to learn consistent local-to-global representations and mitigate the impact of missing peaks.
（2）MS Fusion: Retrieves candidate spectra similar to the input from a database (using precursor mass and spectral similarity as constraints) and fuses their representations with the input spectrum via cross-attention, providing supplementary information for potentially missing peaks.

**Strengths:**

1. PeakNovo creatively combines masked self-distillation (adapted to MS data via peak-level masking) and candidate spectrum fusion (using cross-attention for integration), addressing the key limitation of missing peaks in de novo sequencing—filling gaps in prior work that focused only on single-module optimization.
2. Rigorous and comprehensive empirical validation: Experiments use three diverse datasets (covering low/high-resolution MS data) with non-overlapping training/test peptides, compare against seven baselines, and include ablation studies, sensitivity analyses, and case studies (Pyteomics-annotated spectra) to fully verify performance and module effectiveness.

**Weaknesses:**

1. The paper only validates PeakNovo on three benchmark datasets (Seven-species, Nine-species, and HC-PT) and lacks testing on real-world noisy mass spectrometry (MS) data, such as clinical samples with low peptide abundance or data generated by low-sensitivity MS instruments. Real-world MS data often has more complex missing peak patterns (e.g., overlapping peaks caused by sample impurities or background noise) than the artificially simulated missing peaks in benchmark experiments.
2. The paper does not report key computational efficiency metrics of the MS Fusion module, including inference time per spectrum and memory usage during batch processing. In large-scale proteomics studies, researchers often need to analyze datasets containing millions of spectra, and computational efficiency is a critical factor for the practical adoption of a method. For example, if PeakNovo requires significantly more time or memory than baseline models like CasanovoV2, it may not be feasible for large-scale data analysis.
3. Although the paper states that PeakNovo outperforms baselines in post-translational modification (PTM) identification, it does not specify which types of PTMs (e.g., phosphorylation, acetylation, methylation) were included in the experiments. Different PTMs have distinct chemical properties and abundances in biological samples, leading to varying detection difficulties—for instance, phosphorylation (a low-abundance but biologically critical PTM) is often harder to identify than acetylation. Additionally, the paper does not mention whether PeakNovo can handle complex PTM scenarios, such as multiple modifications on a single peptide.

**Questions:**

1. Can you supplement experimental results of PeakNovo on real-world clinical samples (e.g., tumor tissue-derived peptide samples) with naturally occurring missing peaks (rather than artificially dropped peaks)? Such results would help verify whether PeakNovo’s robustness to missing peaks translates to practical clinical proteomics scenarios, where MS data quality is often compromised by sample complexity?
2. Could you provide specific computational efficiency data for PeakNovo, including inference time per spectrum and memory usage when processing batches of 32 (consistent with the training batch size), and compare these metrics with baseline models like CasanovoV2? This information is essential to assess whether PeakNovo can be applied to large-scale MS datasets commonly used in proteomics research.
3. For the PTM identification experiments, please explicitly list all PTM types included in the evaluation (e.g., phosphorylation at serine/threonine, acetylation at lysine) and provide separate performance metrics (precision, recall) for each type. Additionally, can you clarify whether the experiments included peptides with multiple PTMs, and if so, how PeakNovo performs on these complex cases?
4. Why was InstaNovo (a recent Transformer-based de novo sequencing model mentioned in the related work section that achieves strong performance on high-resolution MS data) not included as a baseline for comparison? If InstaNovo is added to the baseline set, does PeakNovo still maintain its performance advantages in terms of peptide-level precision and AUC?
5. In the Masked Self-distillation module, you used a combination of random masking and contiguous neighborhood masking to simulate missing peaks. Have you conducted comparative experiments with other masking strategies, such as intensity-based masking (masking low-intensity peaks that are more likely to be missing in real MS data)? If yes, what were the results; if no, why was the current masking strategy chosen over alternatives?
6. The paper mentions that the MS Fusion module retrieves candidate spectra using MatchMS with an initial mass tolerance window of ±20 Da, expanding by 10 Da increments if fewer than k candidates are found. Have you tested the impact of different initial mass tolerance windows (e.g., ±10 Da, ±30 Da) on PeakNovo’s performance? Additionally, what criteria were used to determine the optimal value of k (the number of candidate spectra), and how does varying k affect the model’s robustness to missing peak?

---

### Official Review · Reviewer_ab9X · 2025-10-29

**Soundness:** 3
**Presentation:** 3
**Contribution:** 2
**Rating:** 4
**Confidence:** 3

**Summary:**

This paper proposes to use masked self-distillation (MSD) and MS fusion to improve the robustness in de novo peptide sequencing with missing peak condition. The MSD is used to train the encoder to learn consistent representations from local to global view through student and teacher branches. The MS fusion selects similar candidate spectra and fuses them with the input representation to supplement missing fragment information. The approach is tested on three benchmark datasets and compared with several baselines. Results shows that the proposed approach reaches higher accuracy with various missing-peak conditions.

**Strengths:**

1. The paper investigates and address the negative impact from missing peaks, which is novel in biological and medical domains.

2. The paper has two major contributions, where MSD targets robusness improvement and MS Fusion realizes information augumentation. The combination is novel and techincally sound.

3. The experiment is comprehensive, spanning multiple datasets and baselines. The ablation study in Table 2 is also convincing.

**Weaknesses:**

1. Even though combining MSD and spectral fusion is novel, each component is conceptually adapted from existing idea, making the contribution of the paper application synergy, rather than methodological innovation.

2. The paper needs to have more detailed analysis on the computational cost, especially during the inference time.

3. While empirical improvements are apparent, the paper lacks theoretical grounding or embedding-space analysis for the robustness claim.

**Questions:**

1. The paper claim that masked self-distillation yields robust and invariant representations.
Could you quantify or visualize this invariance (e.g., using feature similarity, variance analysis, or t-SNE/UMAP plots comparing masked vs. full spectra)?

2. How sensitive is PeakNovo’s performance to the quality or completeness of the external spectral database used for retrieval? What happens if the database is domain-mismatched or contains noisy spectra?

3. The cross-attention over multiple retrieved spectra likely increases memory and compute cost. Can you report the inference time and GPU memory usage compared to baseline models such as CasanovoV2 or SearchNovo?

4. Please clarify the training cost and number of parameters compared to baselines. Does the performance gain justify the increased complexity?

---

### Official Review · Reviewer_eELi · 2025-10-30

**Soundness:** 2
**Presentation:** 2
**Contribution:** 2
**Rating:** 4
**Confidence:** 3

**Summary:**

The paper focuses on the task of de novo peptide sequencing. Because mass spectrometry (MS) data often contain missing peaks, the authors propose a masked self-distillation approach to learn robust feature representations. During training, a teacher model—updated via an exponential moving average—is used to guide a student model. The teacher receives the original MS data as input, while the student processes a masked version of the same data. The student is trained to reconstruct the masked features using the teacher’s output as supervision. Furthermore, the method integrates external MS data retrieved from an existing dataset to enhance feature fusion.

**Strengths:**

- The paper is well-written and easy to follow.
- The motivation is strong, addressing feature learning from missing spectra in MS data is a challenging problem.
- The proposed method demonstrates strong performance in experiments compared to various baseline approaches.

**Weaknesses:**

The contributions of the paper are somewhat unclear for several reasons:

- Masked Self-Distillation:
The proposed use of masked self-distillation appears to be directly inspired by DINO, a well-established method in self-supervised learning. If the approach represents a straightforward adaptation without substantial innovation specifically tailored to the domain of mass spectrometry or peptide sequencing, the novelty of the contribution may be limited.

- Feature Fusion from Retrieved MS Data:
One of the fundamental motivations behind de novo peptide sequencing is to eliminate reliance on external spectral libraries. However, the proposed method retrieves similar MS data from an external dataset for feature fusion, which seems to reintroduce the dependency that de novo approaches seek to avoid. This raises concerns about the consistency between the stated goals and the actual design of the method.

- Justification for Method Choice:
The paper does not clearly justify the selection of masked self-distillation over alternative self-supervised learning approaches, such as Masked Autoencoders or dense contrastive methods. Providing a rationale or comparative analysis would strengthen the argument for the method’s relevance and suitability to the task.

While the empirical results are promising, the proposed approach lacks a clear motivating rationale and offers limited methodological novelty. The paper would benefit from a more explicit justification of design choices and a deeper discussion of how the proposed method advances de novo peptide sequencing beyond existing techniques.

**Questions:**

The dependency on external MS data appears to be at odds with the goal of de novo peptide sequencing, which aims to operate independently of spectral libraries. Could the authors clarify whether this dependency is limited, and how it aligns with the de novo setting?

---

### Author Response · Authors · 2025-12-03
**Many thanks to all the reviewers**

Dear reviewers:

Thank you very much to all reviewers for their sincere and insightful comments and reviews!

Due to extensive experiments and revisions to the paper, a significant amount of time is required to complete the overall work.

Unfortunately, we were unable to complete all the experiments and paper revision due to the limited time for rebuttal. We will submit the PeakNovo to another conference. Meanwhile, in the new version, we will try our best to address the concerns of all reviewers.

Thank you again for valuable suggestions from all reviewers !

Best regards,

Authors

---

### Meta-Review · Area_Chair_5JCQ · 2025-12-22

**Summary:**

- Lacking novelty / methodological contribution/Lacking theoretical grounding: the approach seems to be a
straightforward adaptation of existing methods to mass spectrometry, but the novelty of the contribution is limited.

- Problems concerning the analysis on the computational cost.

- Concerns about the true "de-novo" nature of this approach.

**Reviewer Concerns:**

Essentially, all points of criticism are still valid. In their (very short) rebuttal, the authors say that they do not have enough time to improve the paper now, and that they will resubmit t to another conference...

**Reviewer Scores:**

I think, none of the reviewers have changed their scores.

---

### Decision · Program_Chairs · 2026-01-26

Reject